# Development and Characterization of High Environmentally Friendly Composites of Bio-Based Polyamide 1010 with Enhanced Fire Retardancy Properties by Expandable Graphite

**DOI:** 10.3390/polym14091843

**Published:** 2022-04-30

**Authors:** David Marset, Eduardo Fages, Eloi Gonga, Juan Ivorra-Martinez, Lourdes Sánchez-Nacher, Luis Quiles-Carrillo

**Affiliations:** 1Textile Industry Research Association (AITEX), Plaza Emilio Sala 1, 03801 Alcoy, Spain; dmarset@aitex.es (D.M.); efages@aitex.es (E.F.); egonga@aitex.es (E.G.); 2Technological Institute of Materials (ITM), Universitat Politècnica de València (UPV), Plaza Ferrándiz y Carbonell 1, 03801 Alcoy, Spain; lsanchez@mcm.upv.es (L.S.-N.); luiquic1@epsa.upv.es (L.Q.-C.)

**Keywords:** PA1010, expandable graphite, mechanical properties, flame retardant, environmentally friendly composites

## Abstract

Bio-based polyamide 1010 was melt-compounded with different percentages (2.5 to 10.0 wt.%) of expandable graphite (EGr) as an environmentally friendly solution to improve the flame retardancy properties. The mechanical, morphological, thermal and fire retardancy properties (among others) are analysed. The novelty of the article lies in the use of fully removable polyamide. The effect of the incorporation of EGr in the properties of this polymer was analysed and characterised. The incorporation of EGr into the PA1010 matrix led to very promising results. Mechanically, the EGr provided increased stiffness and a tensile strength up to 7.5 wt.%, verifying good mechanical performance. The DMTA results also show how the incorporation of EGr in the PA1010 matrix clearly increases the stiffness of the composites over the entire temperature range analysed. In terms of physical properties, water absorption of PA1010 was reduced particularly in the 10% EGr, which reduces the water absorption of PA1010 by 20%. In terms of flame retardant properties, with the incorporation of EGr, a significant reduction in the heat release rate (HRR) values as the concentration of the additive increases and a reduction in the maximum peak heat release rate (pHRR) can be observed for all compounds. In particular, it goes from 934 kW/m^2^ for neat polyamide to a value of 374 kW/m^2^ with 10% EGr. Finally, an improvement in the UL-94 rating of the 7.5 and 10% EGr composites was also observed, going from V-2 in the PA to V-1 in these composites.

## 1. Introduction

In the last decades, social awareness in the search for new environmentally friendly materials has been widely supported by the scientific community and the producing industries. These trends have been boosted by the 2030 sustainable development agenda, where the United Nations has proposed a number of measures to increase sustainability and improve the exploitation of natural resources [1]. One of the most committed industries to the creation of new environmentally friendly materials and additives has been the plastics industry, as polymers are nowadays a basic aspect of our daily lives. They are widely used in different essential and technical sectors of society, such as the production of automobiles, household appliances, packaging and medicine [2,3,4]. This predisposition in the search for highly eco-efficient materials far removed from dependence on oil brings into play polymeric materials of biological and renewable origin [5]. These bio-based materials can be obtained from a variety of natural resources, in the vast majority of cases maintaining very similar properties to their petrochemical counterparts. Thanks to these new trends, more than 2.62 million tons of bio-based polymers are expected to be produced by 2023, a figure that will continue to grow steadily until 2030 [6].

At present, among those biological materials we can find a wide variety of polyamides. Until recently, polyamides were linear and semi-crystalline polymers obtained from petroleum; however, thanks to these new trends, several polyamides have been obtained partly or entirely from bio-based building blocks [7]. In this context, the use of monomers of biological origin such as sebacic acid, 1,4-diaminobutane (putrescine), dicarboxylic acid or 1,5-diaminopemthane (cadaverine) can be used to obtain different types of polyamides with very similar properties to those of their petrochemical counterparts [8,9,10]. In particular, polyamide 1010 (PA1010) can be synthesized from an all-natural source such as castor oil, which plays an important role in the production of bio-based polyamides [11,12]. This is because both carboxylic acid and sebacic acid are obtained from castor oil, resulting in a fully bio-based polyamide [13]. This type of polyamide is particularly useful in engineering applications requiring good heat resistance and flexibility [14]. Despite the excellent properties and applications of polyamides in many engineering sectors, they are inherently flammable. This characteristic limits their use and means that other solutions or additives are required to improve their flame retardant properties [15]. 

Halogenated flame retardant additives have been used in industry for decades with very positive results and performance. Nevertheless, new trends and environmental standards have limited their use, leading to a search for more environmentally friendly non-halogenated flame retardants additives [16]. This problem has generated a great deal of interest in the search for and development of flame retardant (FR) additives that are attractive from an ecological point of view [17]. In recent years, a large number of halogen-free flame retardant alternatives have emerged, facilitating their incorporation into conventional and bio-based polymers. In this block of new FRs, one can find some based on metal hydroxides such as magnesium hydroxide (MH), aluminium hydroxide (AH) or zinc borate (ZB) [18]. Additives such as ammonium polyphosphate (APP) or antimony trioxide (Sb_2_O_3_) have a high capacity in polyolefins [19,20], while those based on phosphorus, such as aluminium hypophosphite (AlHP) or titanium oxide (TiO_2_), have a wide application in general industry [21,22]. However, on many occasions, the incorporation of this type of additive generates a reduction of the mechanical properties of the polymeric matrix, causing certain limitations [23]. Other natural alternatives, such as halloysite nanotubes (HNT), have attracted research attention as fillers for polymer nanocomposites due to their improved flame retardant properties, without losing a large amount of mechanical performance [24,25]. On the other hand, it has been reported that expandable graphite (EGr) is an efficient alternative with a view to improving the flame retardant properties of the material, improving the main problem that polyamides have in terms of heat and flame generation, trying to find a balance of mechanical properties [26,27].

EGr is an intumescent flame retardant additive, highly effective in polymeric materials as it is able to exfoliate and expand at the C-axis of the crystal structure by hundreds of times due to the intercalation agent generating gas via a blowing reaction [28,29]. When this additive is heated above 200 °C, it expands rapidly, becoming a swollen and multi-voided “worm-like” structure. This process results in a large amount of carbonised residues covering the surface of the polymer [30]. In addition, due to its structure and composition, chemical compounds such as CO_2_, H_2_O and SO_2_ are released during its expansion, which dilutes the concentration of flammable gasses released in the flame area [31]. Thanks to the expansion of the graphite layers, this additive is able to consume a large amount of heat, which reduces the combustion heat and burning rate [32]. Several authors have demonstrated that EGr is an effective additive, acting both as a foaming agent and as a carbonizing agent [33]. In recent years, it has proven to be a flame retardant additive with excellent properties when incorporated into materials such as polyethylene (PE) [34,35], polystyrene (PS) [36], ABS [31] or even in conventional polyamides such as PA6 [27,37]. In this context, it has been reported that polymers containing EGr have proven to be particularly efficient in fully developed fires but sometimes show less efficient properties with regard to flammability and self-extinguishing tests [27].

The main objective of this work is to obtain, analyse and characterize polymeric composites with flame retardant capabilities of high environmental efficiency via injection moulding processes introducing different proportions of expandable graphite (EGr) in a PA1010 matrix of totally renewable origin. The main idea has been to evaluate how the incorporation of different percentages of expanded graphite affects the mechanical, thermal and thermo-mechanical properties, as well as to deeply analyse the improvements in terms of flame retardant properties that this additive provides. Currently, expandable graphite is used in an eco-efficient way as a flame retardant, but the novelty and distinctiveness of this work lies in evaluating its use in a fully renewable polyamide matrix, with the aim of analysing how its incorporation affects mechanical, thermal, thermo-mechanical, morphological and fire retardant properties, among others. In addition to the search for a highly efficient compound, in the vast majority of the research mentioned above, only the fire properties of this additive are evaluated. This is intended to provide a wealth of information on this type of high-performance mixture for applications where mechanical properties, efficiency and fire restrictions are essential.

## 2. Materials and Methods

### 2.1. Materials

NaturePlast (Ifs, France) supplied the renewable PA1010 as NP BioPA1010-201. The manufacturer supplies this homopolyamide in pellets as fully bio-based (100% renewable content) and medium-viscosity grade for injection moulding applications. It presents a density of 1.05 g/cm^3^ and a viscosity number (VN) of 160 cm^3^/g. Expandable graphite (C_24_(HSO_4_) (H_2_SO_4_)) was supplied by Sigma Aldrich (Saint-Quentin-Fallavier, France) with CAS number 808121 (Figure 1). 

### 2.2. Sample Preparation

To avoid moisture problems during the processing of the materials, PA1010 was dried at 60 °C for 36 h in a dehumidifying dryer, model MDEO from Industrial Marsé (Barcelona, Spain), to remove any residual moisture prior to processing. PA1010 and the expandable graphite were mechanically pre-homogenized in a zipper bag according to the compositions in Table 1. The selection of the EGr loading was established taking into account the values used in other similar works, where between 5 and 30% of EGr is incorporated [26,38]. In this work, the compositions were selected to evaluate a linear range between 2.5 and 10% of additive, trying to find a balance between mechanical properties and flame retardancy. 

The extrusion was carried out with a twin-screw co-rotating extruder from Construcciones Mecánicas Dupra, SL (Alicante, Spain), with 25 mm diameter screws with a length/diameter (L/D) ratio of 24. The materials were then fed into the main hopper and the rotation speed during extrusion was 18 rpm. The following temperature profile was set: 210 °C (hopper)–215 °C–220 °C–225 °C (nozzle).

The PA1010/EGr composites were extruded through a round die which produced strands that were finally pelletised with an air knife unit. The residence time was approximately 1 min.

The compounded pellets were, thereafter, shaped into standard samples via injection moulding in a Meteor 270/75 from Mateu and Solé (Barcelona, Spain). Three different shapes were injection-moulded to obtain samples for further characterization. Dog bone-shaped samples of 150 × 10 × 4 mm^3^ as indicated by ISO 527-1:2012 were obtained for tensile tests. Rectangular 80 × 10 × 4 mm^3^ samples were obtained for other characterizations and square 150 × 150 × 5 mm^3^ samples were obtained for fire retardancy essays. The temperature profile in the injection moulding unit was 210 (hopper), 215, 220, and 225 °C (injection nozzle). A clamping force of 75 tons was applied while the cavity filling and cooling times were set to 1 and 20 s, respectively. Moreover, a constant filling time of 1 s was maintained for all samples.

### 2.3. Material Characterization

#### 2.3.1. Mechanical Tests

To obtain the mechanical properties of the composites, tensile tests were carried out on an ELIB 50 universal testing machine from S.A.E. Ibertest (Madrid, Spain) using injection moulded samples in the shape of a dog bone according to ISO 527-1:2012. A 5-kN load cell was used and the cross-head speed was set to 5 mm/min. In addition, shore hardness was measured in a 676-D durometer from J. Bot Instruments (Barcelona, Spain), using the D-scale, on injection-moulded samples with dimensions 80 × 10 × 4 mm^3^, according to ISO 868:2003. Finally, toughness was also studied on injection-moulded rectangular samples with dimensions of 80 × 10 × 4 mm^3^ using the Charpy impact test with a 1-J pendulum from Metrotec S.A. (San Sebastián, Spain) on notched samples (0.25 mm radius v-notch), following the specifications of ISO 179-1:2010. All tests were performed at room temperature (25 °C), and at least 5 samples of each material were tested and their values averaged.

#### 2.3.2. Morphology

The morphology of the fracture surfaces of the PA1010/EGr composites, obtained from the impact tests, was observed via field emission scanning electron microscopy (FESEM) in a ZEISS ULTRA 55 microscope from Oxford Instruments (Abingdon, UK), working at an acceleration voltage of 2 kV. Before placing the samples in the vacuum chamber, they were sputtered with a gold-palladium alloy in an EMITECH sputter coating SC7620 model from Quorum Technologies, Ltd. (East Sussex, UK). 

#### 2.3.3. Thermal Analysis

The main thermal transitions, PA1010/EGr composites were obtained via differential scanning calorimetry (DSC) in a Mettler-Toledo 821 calorimeter (Schwerzenbach, Switzerland). An average sample weight ranging from 5 to 7 mg was subjected to the following three-stage dynamic thermal cycle: first heating from 30 °C to 250 °C followed by cooling to 0 °C and a second heating to 300 °C. Heating and cooling rates were set to 10 °C/min. All tests were run in nitrogen atmosphere with a flow rate of 66 mL/min using standard sealed aluminium crucibles (40 μL). The degree of crystallinity ($\chi_{c}$) was determined following Equation (1): (1)$$\chi_{c}\left(\%\right)=\left[\frac{\Delta H_{m}}{\Delta H_{m}^{0}\cdot \left(1-w\right)}\right]\cdot 100$$
where $\Delta H_{m}$ (J/g) stands for the melting enthalpy of the sample, $\Delta H_{m}^{0}$ (J/g) represents the theoretical melting enthalpy of a fully crystalline PA1010, that is, 244.0 J/g [39], and w corresponds to the weight fraction of different fibres in the formulation.

Thermogravimetric analysis (TGA) was performed in a LINSEIS TGA 1000 (Selb, Germany). Samples with an average weight between 15 and 25 mg were placed in standard alumina crucibles of 70 µL and subjected to a heating program from 30 °C to 700 °C at a heating rate of 10 °C/min in air atmosphere. The first derivative thermogravimetric curves (DTG) were also determined, expressing the weight loss rate as the function of time. All tests were carried out in triplicate.

#### 2.3.4. Thermomechanical Characterization

With the aim of obtaining more information about the samples, a DMA1 dynamic analyser from Mettler-Toledo (Schwerzenbach, Switzerland) was used for dynamic mechanical thermal analysis (DMTA). Injection-melded samples with dimensions of 20 × 6 × 2.7 mm^3^ were subjected to a dynamic temperature sweep from −80 °C to 150 °C at a constant heating rate of 2 °C/min, a frequency of 1 Hz and a maximum cantilever deflection of 10 µm. All tests were carried out working in single cantilever flexural conditions.

#### 2.3.5. Colour Measurements

To analyse the colour change of the samples with the addition of expandable graphite, a Konica CM-3600d Colorflex-DIFF2 colourimeter, from Hunter Associates Laboratory, Inc. (Reston, VA, USA) was used for colour measurement in CIELab and CIELCh scales. CIELab colour indexes (L*, a*, and b*) were measured according to the following criteria: L* is the lightness and changes from 0 to 100; a* stands for the green (a* < 0) to red (a* > 0) colour coordinate, while b*, represents the blue (b* < 0) to yellow (b* > 0) colour coordinate. CIELCh colour indexes (L*, C* and h*) are according to the following criteria: L* is the lightness ranged from 0 to 100, C* the colour tone and the h* represents the hue angle (0 = red, 90 = yellow, 180 = green and 270 = blue). Measurements were done in quintuplicate.

#### 2.3.6. Wetting Characterization

Contact angle measurements were carried out with an EasyDrop Standard goniometer model FM140 (KRÜSS GmbH, Hamburg, Deutschland) which is equipped with a video capture kit from Drop Shape Analysis SW21 from KRÜSS (Bristol, United Kingdom) coupled with the DSA1 (v1.9) analysis software from the same company. Double distilled water was used as test liquid. The wetting properties were evaluated on the surface of rectangular 80 × 10 × 4 mm^3^ samples. At least 12 measurements of the water contact angle were collected and averaged.

#### 2.3.7. Water Uptake Characterization

The evolution of water absorption was studied using rectangular samples of 4 × 10 × 80 mm^3^, which were immersed in distilled water at 23 ± 1 °C. The samples were taken out and weighed weekly using an analytical balance with a precision of ±0.1 mg, after removing the residual water with a dry cloth. The evolution of the water absorption was followed for a period of 15 weeks. Measurements were performed in triplicate.

#### 2.3.8. Cone Calorimeter Test (CCT)

The cone calorimeter model was 82121 (FIRE Ltd., Surrey, UK) and the tests were performed according to ISO 5660 standard procedures. The dimensions of the samples were 100 × 100 × 5 mm^3^. Each sample was wrapped in aluminium foil (0.0025 to 0.04 mm thick) and horizontally exposed to an external heat flux of 50 kW/m^2^, 25 mm conical distance and 20 min test time.

#### 2.3.9. Limiting Oxygen Index (LOI) and UL94

LOI was carried out in an 82121 model (FIRE Ltd., Surrey, UK) according to the standard oxygen index test stated in the UNE-EN ISO 4589-2 norm. Type I test pieces and the ignition procedure (A) related only to the upper surface were used. Prior to the test, the specimens were conditioned at 23 °C and 50% relative humidity for 24 h. The size of the samples used was 150 × 10 × 4 mm^3^. Three samples were studied using the LOI test.

The UL-94 horizontal burn tests were carried out following the testing procedure UL 94:2006; EN 60695-11-10:1999/A1:2003 with a test specimen bar that was 125 mm long, 13 mm wide and about 5 mm thick.

## 3. Results

### 3.1. Mechanical Properties of PA1010/EGr Composites

The mechanical behaviour of PA1010 composites with an increasing amount of expandable graphite provides information on the possible applications in which PA1010/EGr composites with fire retardant properties can be used. Table 2 shows the most relevant results after the mechanical characterization such as the elastic modulus (E), the maximum tensile strength (σ_max_) and elongation at break (ε_b_), shore D hardness and the impact strength measured by means of the Charpy method. In this work, the compounds were prepared in a twin-screw extruder in a process known as melt blending or mechanical mixing due to it being the most economically attractive mixing method for an industrial scale-up and that it avoids the use of organic solvents in the process [40]. The use of graphite-type additives often leads to the formation of agglomerates; in some works, this phenomenon is reduced with the aid of ultrasounds to improve particle dispersion within a solvent prior to the mixing process [41,42]. Another technique employed to achieve good dispersion is the use of an in situ polymerization process [43]. 

With regard the tensile modulus, the incorporation of EGr to obtain composites allowed the stiffness to be increased from 1701 MPa of the PA1010 up to 2164 MPa for the PA1010/10EGr composite. The stiffening effect of polyamide composites after the incorporation of different carbon particles is a common phenomenon as observed by Faridirad et al. when analysing the behaviour polyamide composites with different carbon nanoparticles [44]. 

PA1010 resulted in a tensile strength of 45.5 MPa; other authors such as Nikiforov et al. reported similar values for this kind of unmodified polyamide [45]. Depending on the amount of EGr considered, different results were obtained. The tensile strength for the PA1010/2.5EGr composite was 50.5 MPa, showing a clear reinforcing effect. Yu et al. was also able reinforce polyamide 6 with the incorporation of exfoliated graphene up to a proportion of 0.3 wt.%; with a higher amount of graphene the proper dispersion was not acquired, promoting a decrease in the mechanical properties [46]. In this work, the difficulty in achieving a good dispersion for the 5 and 7.5 wt.% composites gave a tensile strength of 48.4 MPa and 46.5 MPa, respectively. Despite the reduction obtained compared with the PA1010/2.5EGr, a higher tensile strength than that obtained for the unmodified polymer was observed. In contrast, the composite with the highest amount of EGr exhibited a tensile strength of 42.9 MPa, which is lower than the neat PA1010, showing that the effect of the agglomerates had more impact than the reinforcement obtained with the EGr. This behaviour, in which the addition of carbon nanomaterials reduces the tensile stress, has been reported by authors such as Dittrich et al. [47]. After analysing different PP (polypropylene) composites, a reduction of the mechanical properties was observed mainly due to the difficulty of achieving a good filler dispersion in the polymer matrix. This mechanical behaviour was improved by functionalizing the additives to improve the interaction with PP, resulting in an improvement of the tensile strength. Despite being able to increase the stiffness and strength of the material, the elongation showed a decreasing trend as a function of the amount of expandable graphite considered. The elongation at break obtained was 237.4% for PA1010 and only 5.7% for the sample with 10 wt.% of EGr. This same behaviour was reported by W. He et al. when introducing different flame retardant additives in a PA6 polymeric matrix [48]. There are different reasons for the reduced elongation of the composites with the introduction of expandable graphite as proposed by Sever et al. [49]. The introduction of the expandable graphite leads to a reduction of the movement of the polymer chains and also the formation of aggregates due to poor dispersion of the additive, promoting the reduction of the ductile properties. The hardness of the composites as a function of the amount of EGr follows a similar trend to the tensile modulus, resulting in the hardness of PA1010 going from 74.2 Shore D to 76.1 Shore D. As mentioned above, the introduction of EGr particles results in a lower mobility of the polymeric chains, thus providing a reinforcement function. Similar results were described by Piana et al. after incorporating expandable graphite into a polyurethane polymeric matrix [50]. The impact strength of the polyamide showed a highly ductile behaviour with 9.5 kJ/m^2^. The incorporation of graphite resulted in a reduction of the energy absorption capacity during impact. In this sense the incorporation of 2.5 wt.% EGr results in 5.1 kJ/m^2^ and this value was reduced to 2.0 kJ/m^2^ with the incorporation of 10 wt.% EGr following the same trend that occurs with the elongation at break of the composites. The presence of EGr has been reported as the origin of agglomerate formation due to the dispersion problems that arise, leading to stress concentrators and micro-cracks that resulted in the loss of ductile properties [51]. 

### 3.2. Morphology of PA1010/EGr Composites

The surface of the fractured specimens in the Charpy impact test was analysed using FESEM (field emission scanning electron microscopy) so that the behaviour of the composites during fracture could be assessed. Figure 2a shows the morphology of PA1010 during fracture, which shows signs of having undergone plastic deformation as inferred from the presence of a rough surface [12]. The incorporation of EGr makes it possible to observe the presence of particles embedded within the polymeric matrix in the form of agglomerates in the 5 wt.% sample (Figure 2c). As the mechanical results suggest, the expanded graphite particles could not be properly dispersed within the polymeric matrix, a clear example of which is Figure 2e showing the PA1010/10EGr composite, in which a large agglomerate can be observed. Kim et al. also reported the presence of agglomerates by analysing the morphology in polyamide 66 composites with graphene [52]. As a result of the incorporation of graphite, an additional modification could be appreciated. The morphology of the polyamide changed from a rough surface characteristic of plastic deformation to a smooth surface of a polymer with a low plastic deformation capacity. This modification is linked with the reduction of the ductile properties of the polyamide as a consequence of the incorporation of graphite [53]. 

The correct interaction between the polymer and the filler can be analysed via the presence of deboned particles in the fracture surface, as suggested by Kodal et al. [54]. In this case, there are no clear signs that the graphite particles were deboned during breakage, suggesting that there is a good interaction between the two components, which allowed the reinforcing effect of the composites that improved the tensile strength of PA1010 at fire additive ratios between 2.5 wt.% and 7.5 wt.%.

### 3.3. Thermal Properties of PA1010/EGr Composites

Figure 3 shows the heating thermograms obtained via differential scanning calorimetry (DSC) of the PA1010/EGr composites; additionally, the most relevant parameters obtained during the analysis are shown in Table 3. The only thermal transitions that could be identified via dynamic DSC in the considered temperature range are the melting peaks of the PA1010 between 190 °C and 210 °C due to its semicrystalline structure. Different melting behaviour between the first heating cycle and the second (Figure 3a,b) were observed. During the first cycle, samples were obtained from a fragment of a tensile specimen made via an injection process. As a result, PA1010 was cooled down to room temperature in only 20 s. In contrast, the second heating cycle was performed after controlled cooling at 10 °C/min. The crystalline structure obtained during the injection moulding process showed a single melting peak obtained around 203 °C, while during the second heating cycle an additional melting peak appears at 192 °C. The same behaviour between both thermal cycles in PA1010 could also be observed by Nishitani et al. [55]. The presence of a single melting peak is promoted by a single crystalline form, whereas the presence of several melting peaks is common in short-chain polyamides due to polymorphism where different crystal structures are formed as a consequence of the crystallization conditions that took place [13]. For the PA1010, different crystal forms have been reported such as the triclinic alpha-crystal structure and the pseudo γ-hexagonal crystal [56]. In addition, the incorporation of expandable graphite had no effect on the melting temperatures observed in both heating cycles, as reported by Yan et al. and Nishitani et al. [39,55].

Regarding the degree of crystallinity as a function of the heating cycle considered, samples acquired a higher degree of crystallinity during the second cycle due to a controlled cooling process at 10 °C/min. Frihi et al. showed that slower cooling rates allow the polymer chains to rearrange to achieve a higher degree of crystallinity [57]. The increase in crystallinity with the amount of EGr could be observed in both cycles. This situation suggests that graphite is acting as a nucleating agent, with a degree of crystallinity of 21.7% for the PA1010/7.5EGr in the second heating cycle, while the PA1010 presented a value of 19.1%. The presence of higher amounts of graphite did not provide an enhancement of the degree of crystallinity due to aggregate formation hindering the crystallization process, thus avoiding the formation of large crystalline regions [58]. 

Thermal stability of the PA1010/EGr composites was assessed by means of thermogravimetric analysis, and the diagrams are presented in Figure 4, while Table 4 gathers the main thermal parameters related to this test. In all cases, thermal degradation of the composites took place in a single step, in which the polyamide was degraded due to a chain scission of the N-alkylamide bond [59]. 

PA1010 presented a high thermal stability, presenting an onset degradation temperature of 409 °C taking as a criterion a 5% mass loss (T_5%_). This parameter experienced a slight difference resulting in the highest value being obtained for the composite with a 7.5 wt.% EGr with a T_5%_ of 412 °C. The maximum degradation rate temperature (T_deg_) was slightly modified by the presence of graphite; the measured values were between 438 °C and 444 °C with the lowest T_deg_ value obtained for composites with a higher amount of EGr. This modification suggests that poor dispersion of the EGr in the composites promoted a small decrease in thermal stability. In this sense, the study of Hassouna et al. showed that for a PLA composite with 3 wt.% EGr, the degradation temperature of the polymer could be enhanced via a good dispersion of the additive [60]. Graphite is characterized by high thermal stability resulting in only 10% mass loss after thermogravimetry testing at 700 °C for expandable graphite, as reported by Chen et al. [61]. As a result, the residual weight obtained during the test increased depending on the amount of graphite of each formulation. While PA1010 showed a residue of 0.5%, for the composite with 10 wt.% EGr the amount of residual weight increased up to 10.5%. The formation of ash plays an important role in fire conditions since the graphite at temperature range between 280 °C and 438 °C expands giving rise to a protective layer with high porosity which delays the ignition time [33]. 

### 3.4. Thermomechanical Characterization of PA1010/EGr Composites 

The incorporation of expandable graphite into the PA1010 matrix was studied under dynamic conditions via dynamic mechanical thermal analysis (DMTA). The results are presented in Figure 5, where Figure 5a shows the evolution of the storage modulus (E’) as a function of temperature, while Figure 5b shows the evolution of the dynamic damping factor (tan δ). Table 5 shows the main thermo-mechanical properties of all samples.

In general terms, the storage modulus of the samples follows an increasing trend, as previously observed in the tensile modulus, which increases as a function of the amount of EGr in the composite. At −90 °C, pure PA1010 offers a storage modulus of 1285 MPa, while the composite with 10% EGr showed an E’ value of 2500 MPa. As the test temperature increases, the values decrease significantly, so that at 125 °C it goes from 125 MPa (PA1010) up to 350 MPa for PA1010/10EGr. In the temperature range between 30 °C and 60 °C, the storage modulus decreased progressively to values of 100–300 MPa. This drop in mechanical stiffness is attributed to the alpha transition region (α) of PA1010, in which the amorphous phase of the biopolymer changes from a glassy to a rubbery state [62]. As can be seen, the addition of EGr contributes positively to the increase of the stiffness of the material, both below and above the characteristic T_g_ of the base material. This phenomenon is closely related to the inhibition of plastic deformation observed in tensile tests. Other authors have reported similar increases, where the stiffness of the composites is significantly increased over the entire studied temperature range [6]. Regarding the dynamic behaviour of PA1010, Figure 5b shows the two characteristic peaks associated with the different relaxations of its structure. The first of these peaks is located around −75 °C for all compounds. This peak is the β relaxation, typically attributed to non-hydrogen-bonded amide groups and more specifically to water bonded to carbonyl groups [63]. Within experimental error, this relaxation seems to be unaffected by the incorporation of different percentages of EGr into the polymer matrix. Secondly, the maximum peak observed in the dynamic damping factor diagram is indicative of the α relaxation, which can be directly attributed to the glass transition temperature (T_g_) of the composites. On the one hand, it can be appreciated how PA1010 shows a glass transition temperature of 49.7 °C [64]. The addition of EGr to the matrix causes a slight increase of the glass transition temperature from 49.7 to 51.7 °C for PA1010 and PA1010/10EGr, respectively. This increase may not be significant; however, this alteration may be attributed to the mobility reduction of the polymer chains due to the EGr introduction [65]. 

Finally, it can be seen how the incorporation of EGr into the PA1010 matrix represents a clear increase in the stiffness of the composites, especially for the mixture with 10% EGr, which doubles the stiffness of the base material over the entire working temperature range.

### 3.5. Colour Measurement and Visual Appearance 

In order to determine how the incorporation of expandable graphite affects the visual appearance and colour difference of the biopolyamide matrix, colour space results in CIELAB (L *, a *, b *) and CIELCh (L*, C*, h*) colour space coordinates are analized. In this context, colorimetric results obtained in the different composites after the fabrication process are shown in Figure 6, while colour coordinates of the CIELab and CIELCh chromatic space are found in Table 6.

The incorporation of the EGr in the mixture results in a noticeable change in the visual appearance and colour change of the original matrix. PA1010 obtains lightness values above 60, while the coordinates a (−2.35) and b (−6.63) indicate that the pure material tends towards blue colours according to the CIELab colour space. These results are similar to those of other authors regarding the shades of this type of polyamide [66]. If the direct incorporation of the EGr is analysed visually, it can be seen that there is no significant change with the increase in the percentage of EGr. However, as it has been already mentioned, there is a clear visual difference between the original matrix and the loaded samples, where in Figure 6, it can be seen that all of them tend towards a colour near black. 

If the values in Table 6 are analysed, it can be seen that the four compositions that have been loaded with EGr tend to have values of a and b close to 0. This value is indicative of very pure black or white colours. In this context, it can be seen that as the amount of EGr in the mixture increases, the luminosity values (L*) decrease, obtaining values below 35.34 for the PA1010/10EGr blend. This drop in the sample’s luminosity indicates that as the loading percentage (EGr) increases, the pieces tend to acquire a more intense black. These values make sense if we assume that the colour of the expanded graphite particles is an intense glossy black (Figure 6). Authors such as Šehić et al. have reported similar results with the incorporation of carbon nanotubes or carbon black in a polyamide matrix [67]. To complement the information reported by the CIELab colour space, information on the CIELCh space has been calculated and added. This method allows information on colour purity and predominance to be obtained from the CIELab space. The CIELCh polar system includes the hue angle (h), which describes the predominant colour perceived by an observer. In this system, 0° = red-purple, 90° = yellow, 180° = blue-green and 270° blue. In addition, the chroma (C) describes the percentage of purity (C = 100) and the weakness of the colour (C = 0) [68]. Table 6 shows how PA1010, with a hue angle of 250°, verifies the tendency towards blue, while the chroma of 7 indicates a very weak colour, hence the whitish-transparent appearance. In contrast, compounds with EGr tend to have hue angles close to 90, indicative of yellow colours. As far as colour purity (C) is concerned, this composition has values of 0, which verifies the black colour seen in the CIELab.

### 3.6. Wetting Properties of the PA1010/EGr Composites

The incorporation of additives focuses on the improvement or modification of certain properties of materials. However, unexpected results are occasionally obtained. In this context, the water contact angle (θ_w_) of neat PA1010 and PA1010/EGr blends was obtained in order to evaluate their wetting properties. Figure 7 shows the contact angles θ_w_, of the different compounds. As it can be observed, all of them are above 75°. High contact angles are indicative of a poor affinity for water, which generates certain advantages in several applications. Contact angles above 65° indicate hydrophobicity, as it could be considered the hydrophilicity threshold [69]. Particularly, it can be seen how the incorporation of EGr implies a clear advantage from a hydrophobic point of view. The contact angle increases as the EGr content is superior in the blend. Neat PA1010 gives a value of 75.8°, while the blend with 10 wt.% of EGr presents a contact angle of 82.3°. The contact angle results of the polyamide are very similar to those reported by other authors, where polyamide 6 exhibits values very close to 76° [70].

The water contact angle is closely related to the surface energy. In this case, the structure of the polyamide is formed by active amide groups, hence its ability to absorb water, due to the formation of hydrogen bonds. This is the reason why neat polyamide presents a reduced contact angle. Nonetheless, the incorporation of EGr significantly improves the hydrophobicity of the blend. This improvement in the hydrophobicity of the materials is related to the change in crystallinity and the interaction between the amide groups and the structure of EGr. Thus, reducing the content of amide groups per volume unit, so the water contact angle suffered an increase as the concentration of the flame retardant additive increased too. Song et al. [71] reported very similar results with the incorporation of basalt fibre in a PA1012 matrix.

### 3.7. Water Uptake Characterization

Polyamide is a material with a clear disadvantage in relation to the water absorption it can take up due to its high polarity. This factor proves to be a disadvantage for certain industries and applications, so the measurement of this property is relevant to assess how it performs in very high humidity conditions. Figure 8 shows the evolution of water absorption of injection moulded parts during 12 weeks of immersion in distilled water.

Generally, the vast majority of fillers and reinforcements added to polymeric matrices generate a variation in the amount of moisture that can be absorbed by the samples due to the fact that the structure of the polymer is usually modified. In the case studied, the incorporation of EGr is a clear advantage from this point of view. Although PA1010 is not extremely prone to moisture absorption due to its high CH_2_/CONH ratio, the sample without EGr absorbed approximately 1.25% by weight of water. These results are very similar to those reported in previous studies [13]. The incorporation of EGr implies values of 1.2%, 1.15%, 1.10% and 1.0% at week 12 for the samples loaded with 2.5, 5, 7.5 and 10% EGr, respectively. In particular, it can be seen that the mixture with 10% EGr reduces the water absorption of PA1010 by 20%. This reduction in water absorption may be related to a higher crystallinity achieved in the parts, as water molecules are absorbed only in the amorphous regions by involving two close amide groups in an accessible region [72]. These results are very similar to those reported by Quiles-Carrillo. et al. [62], who showed how the addition of treated slate fibre to a PA1010 matrix also reduced the water absorption capacity of the composite, also associated with a higher crystallinity and a better coupling of the fibres. Therefore, the incorporation of EGr favours a direct and proportional reduction in the amount of water that the sample can absorb, reducing the problems derived from the water absorption of the polyamide.

### 3.8. Cone Calorimeter Test (CCT)

To evaluate the flame retardant properties of EGr in PA1010 matrix, the cone calorimetric test (CCT) was carried out. The CCT simulates the combustion of polymers in a real fire situation, showing great importance in research and allowing the development of new materials with excellent flame properties [73]. Table 7 shows the main results obtained in this test for PA1010/EGr composites.

The incorporation of EGr into PA1010 results in a clear change in flame retardant properties. Firstly, there is a clear reduction in the heat release rate (HRR) values as the EGr concentration increases. A reduction in the maximum peak heat release rate (pHRR) can be observed for all compounds. In particular, it goes from 934 kW/m^2^ for pure polyamide to a value of 374 kW/m^2^ for the composite with 10% EGr. This reduction is a clear advantage in terms of fire properties, as the amount of heat emitted by the material is reduced by 60% as a maximum value. These results are largely related to the physical carbon barrier generated by the EGr with increasing temperature and the synergistic effect created by this element in the structure of PA1010 [74]. 

Figure 9 shows the evolution of the heat emitted as a function of time, where it can be seen how the incorporation of the EGr notably reduces the maximum heat peaks (pHRR). The sample with 2.5% EGr reduces the maximum emitted heat value by 94 kW/m^2^ and delays this peak by more than 80 s. The compounds with 7.5 and 10% EGr, respectively, obtain very low peaks, close to 400 kW/m^2^, extending the heat emission up to 800 s. This direct reduction in heat released with the incorporation of EGr is largely related to the intumescent behaviour of this element. When heated, EGr rates the expansion of the clutter, producing an intumescent carbon layer that significantly reduces heat emission [75].

The HRR measured using a cone calorimeter is a very important parameter as it expresses the intensity of the fire [76]. This reduction of the peak indicates a lower energy released during the test, verifying the possible application of EGr as a flame retardant additive in certain applications.

Regarding the time to ignition (TTI) of the samples, this value decreases with the addition of EGr to the compound, i.e., there is more ease of ignition and the duration of ignition is also higher. However, these are ignitions that give off a lower amount of heat. In this context, polyamides are materials with high ignition times when compared to other polymers such as polyethylene or polypropylene [77]. However, the incorporation of EGr reduces ignition times by more than 30 s due to intumescence processes, which are very beneficial for generating a compact carbon layer, making combustion more difficult, but reducing the TTI. In addition, the peak heat release time (pHRR) decreases, a factor that is largely related to the decrease in TTI. Finally, if the sustained ignition times (t_sos.inflamability_) are analysed, several changes in the compounds can be seen. The addition of 2.5% EGr reduces this value by 300 s. On the other hand, the incorporation of higher amounts of EGr increases this value. This increase is closely related to what was seen in Figure 9, where the effect of the EGr reduces the HRR value emitted but generates a longer ignition time. Both TTI and t_sos.inflamability_ results are largely related to the intumescence processes of EGr. When this element is heated above 200 °C, it rapidly expands into a swollen, multi-hollow structure, resulting in a large amount of carbonized residues covering the polymer surface [30]. This temperature is below the melting and degradation temperature of the polyamide, which directly generates a small reduction in these times, allowing a significant reduction in the energy released by the system.

Regarding effective heat combustion (EHC), this represents the heat released during combustion per unit mass. Table 7 shows that all the samples have similar values, the samples with 2.5 and 5% EGr being the ones with the highest EHC values, respectively. These values make sense since, although the incorporation of EGr reduces the heat emission significantly, the samples remain in sustained ignition for longer due to the intumescence of the additives. The amount of released heat is significantly reduced, but the ignition of the samples is maintained for a longer time due to the generation of the carbon layers. This reduction is due to the fact that the carbon atoms in the delayed mixtures can hardly be completely transformed into carbon dioxide, as the foamed carbonised structures formed on the surface of the material become a thermal insulation material that prevents combustible gases from feeding the flame. Only when the material absorbs more heat, will this layer break down, allowing combustion to continue [78]. 

Finally, in order to be able to analyse more directly the flame retardant properties of the compounds, Vahabi et al. [79] defined a dimensionless concept called the “Flame Retardancy Index” (FRI), which allows a very simple comparison between the pure polymer and its flame retardant compound. Equation (1) shows the dimensionless concept: (2)$$\mathrm{FRI}=\frac{{\left[\mathrm{THR}\cdot \frac{\mathrm{pHHR}}{\mathrm{TTI}}\right]}_{\mathrm{Neat Polymer}}}{{\left[\mathrm{THR}\cdot \frac{\mathrm{pHHR}}{\mathrm{TTI}}\right]}_{\mathrm{Composite}}}$$

This comparative dimensionless value allows a straightforward evaluation of the information obtained in Table 7, allowing a comparison between the pure polymer and its composites. Following the guidelines of Vahabi’s work, any composite with an FRI value below 1 is a “poor” performer in terms of fire retardancy. Following this premise, the incorporation of EGr gives good results, which is a great improvement in terms of the fire retardant properties of the composites. In particular, very good results are obtained for the composites with 7.5 and 10% EGr, obtaining FRI values of 1.31 and 1.34, respectively.

Regarding the values related to the smoke generated during the CTT test, Table 8 shows the most significant parameters.

In certain applications, the smoke generation and performance of a flame retardant material is a vital parameter in terms of fire safety. As far as smoke generation is concerned, no significant changes are noticeable. It should be noted that for compounds with 2.5, 5 and 7.5% EGr, the total amount of smoke slightly increases. This factor is related to the physical barrier effect of the carbonized layer formed, which prevents the diffusion of oxygen and favours the development of incomplete combustion products.

On the other hand, the quantification and emission of CO_2_ and CO are relevant as they can generate problems in the surrounding environment of the materials in case of fire. In this context, the incorporation of EGr into the PA matrix does not lead to a large increase in CO_2_ emission values, but it does increase the amount of CO emitted for the samples with 7.5 and 10% EGr. This slight increase is strongly related to the generation of carbonized residues during the burning of the EGr. As already mentioned, this layer reduces the heat and mass transfer, reducing the heat emission. Chemical compounds, such as CO_2_, H_2_O and SO_2_, are released during EGr expansion, which dilutes the concentration of flammable gases released in the flame area. The expansion of the graphite layers also consumes a large amount of heat, which reduces the heat of combustion and the rate of combustion [32]. 

Regarding the smoke extinguishing area (SEA), this is a measure of smoke density, and its value should be as low as possible to make it easier for people to escape from a fire situation. The incorporation of EGr means a clear reduction of this value for mixtures with 7.5 and 10% EGr, and the values of PA 1010 are reduced by more than 220 m^2^/kg. This is a clear improvement in terms of fire performance, as the generation of noxious fumes is reduced. Similar values have been reported by other authors with the incorporation of expanded graphite [80].

### 3.9. Limiting Oxygen Index (LOI) and UL94

LOI and UL-94 tests provide a simple and intuitive way to observe the flame retardant properties of polymeric materials. The LOI test aims to determine the minimum percentage of oxygen required in a mixture to maintain the combustion of the sample after ignition. These tests are widely used to evaluate the flame retardant properties of materials, especially for the selection of flame retardant polymer formulations. On the other hand, the UL-94 test allows the evaluation of the flame behaviour of plastic materials, classifying them according to the extinguishing time and if dripping is present. Table 9 shows the LOI and UL-94 values obtained for PA1010 with different concentrations of EGr in their structure.

If the values obtained for the LOI test are analysed, a decrease in the O_2_ value can be seen as the concentration of EGr in the polymer matrix increases. The decrease in % O2 is a negative factor from the point of view of flame retardant properties, as it makes the material easier to ignite. With the exception of the 2.5% EGr mixture, which increases the amount of oxygen needed to maintain ignition by 1%, the 5, 7.5 and 10% EGr mixtures slightly reduce the amount of O_2_ needed. It should be noted that PA1010 has relatively high LOI values. Conventional polymers such as polypropylene and blends of PP with PA usually have values below 20%, which demonstrates the excellent properties of this type of all-natural PA in certain technical applications [26].

It should be noted that the behaviour of the studied samples is very different. For PA1010, PA1010/2.5 EGr and PA1010/5EGr, the samples melt, and the test is stopped by the extension of the flame when it reaches the mark on the specimen. On the other hand, the samples with 7.5 and 10% EGr show carbonisation with intumescence, and the tests are stopped because they maintain an ignition more than 180 s despite not reaching the mark due to the combustion in these cases being much slower. In the same way as in the LOI, in the UL 94 test, the PA 7.5 EGr and PA 10 EGr specimens carbonise, so despite having long ignition times, there is no droplet fall and they are classified as V-1, while the other samples melt with droplet fall, so they are class V-2. 

Figure 10 shows the visual appearance of the samples after the test, verifying the excellent flame properties of the compounds with 7.5 and 10% EGr, respectively. These appearance, verify previously reported results by the EGr incorporation; where the EGr expanded rapidly in carbon layers forming a barrier prevents heat transfer between the flame zone and the combustion matrix, which delays pyrolysis to some extent [81,82].

## 4. Discussion

The results herein presented open up the possibility of obtaining very environmentally friendly composite materials thanks to the exceptional combination of a 100% renewable polyamide and a halogen-free flame retardant additive with high environmental efficiency and good properties, called expandable graphene. The search for halogen-free flame retardant coatings is currently of particular interest due to the environmental problems they cause. In relation to mechanical properties, the PA1010/EGr composites provided increased stiffness, verifying good mechanical performance. In particular, the composite with 7.5% EGr increased the Young’s modulus and hardness of the composites, reaching values of 2085 MPa and 75 shore D, respectively, obtaining very interesting values compared to pure PA1010 (1701 MPa and 74.2 Shore D). These results were verified using FESEM images, where a good interaction between the two components was observed. This interaction allowed the reinforcing effect of the composites that improved the tensile strength of PA1010 at fire additive ratios between 2.5 wt.% and 7.5 wt.%. In relation to thermo-mechanical properties, the incorporation of EGr into the PA1010 matrix represents a clear increase in the stiffness of the composites, especially for the mixture with 10% EGr, which doubles the stiffness of the base material over the entire working temperature range. At −90 °C, pure PA1010 offers a storage modulus of 1285 MPa, while the composite with 10% EGr showed an E’ value of 2500 MPa. As the test temperature increases, the values decrease significantly, so that at 125 °C it goes down to 125 MPa (PA1010) and 350 MPa for PA1010/10EGr. It should be noted that the incorporation of EGr reduces the water absorption capacity of PA 1010, as well as reducing its hydrophilicity. In particular, it can be seen that the mixture with 10% EGr reduces the water absorption of PA1010 by 20%. Finally, in terms of flame retardant properties, the incorporation of this additive generates a reduction in the maximum peak heat release (pHRR) for all compounds. Specifically, it goes from 934 kW/m^2^ for pure polyamide to a value of 374 kW/m^2^ for the composite with 10% EGr. This reduction is a clear advantage in terms of fire properties, as the amount of heat emitted by the material is reduced by 60% as a maximum value. In addition, it is worth noting that the 7.5 and 10% carbonised specimens, despite having long ignition times, do not show droplet fallout. These results place these two compositions as V-1 in the UL-94 tests, while the rest of the samples (including PA1010) are placed as class V-2. Finally, thanks to the balance of mechanical properties, improved water absorption, colour change and excellent fire retardant properties, this type of composite can be used in different industries and applications. In particular, the improvement in fire properties is very relevant in applications where fire safety is crucial. These applications are focused on interior parts in the automotive, railway or even aviation industry or more technical parts such as engine parts or electrical elements.

## 5. Conclusions

The results obtained open up the possibility of obtaining very environmentally friendly composite materials thanks to the exceptional combination of a bio-based polyamide (100%) and a highly environmentally efficient flame retardant additive. The FESEM images together with the results of the mechanical properties show a good interaction between EG and PA1010, showing a reinforcement that improves the tensile strength of the neat polymer up to 7.5 wt.% although the elongation at break and impact energy capacity was reduced. From the thermal point of view, no major differences emerged although the thermo-mechanical properties were improved due to the reinforcing effect of the EGr. From the point of view of the properties of the composites under fire conditions, the incorporation of the expandable graphite improved the values of energy released during combustion. In the UL-94 test, it was possible to observe a reduction in the dripping of the composites during the combustion process, resulting in the PA1010/7.5EGr and PAPA1010/10EGr classified as being V1. The combination of PA1010 and EGr results in highly efficient composites, with a clear improvement from a flame retardant point of view. In addition to this, the physical properties reported show balanced composites, both mechanically and thermally. As a result, highly environmentally efficient composites were obtained with very promising final properties.

## Figures and Tables

**Figure 1 polymers-14-01843-f001:**
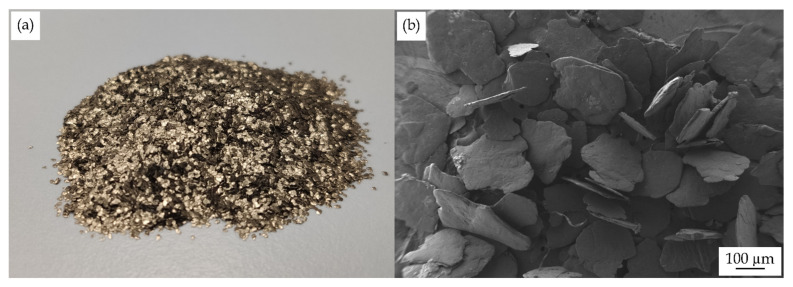
(**a**) Visual aspect and (**b**) field emission scanning electron microscopy (FESEM) images at 30× expandable graphite.

**Figure 2 polymers-14-01843-f002:**
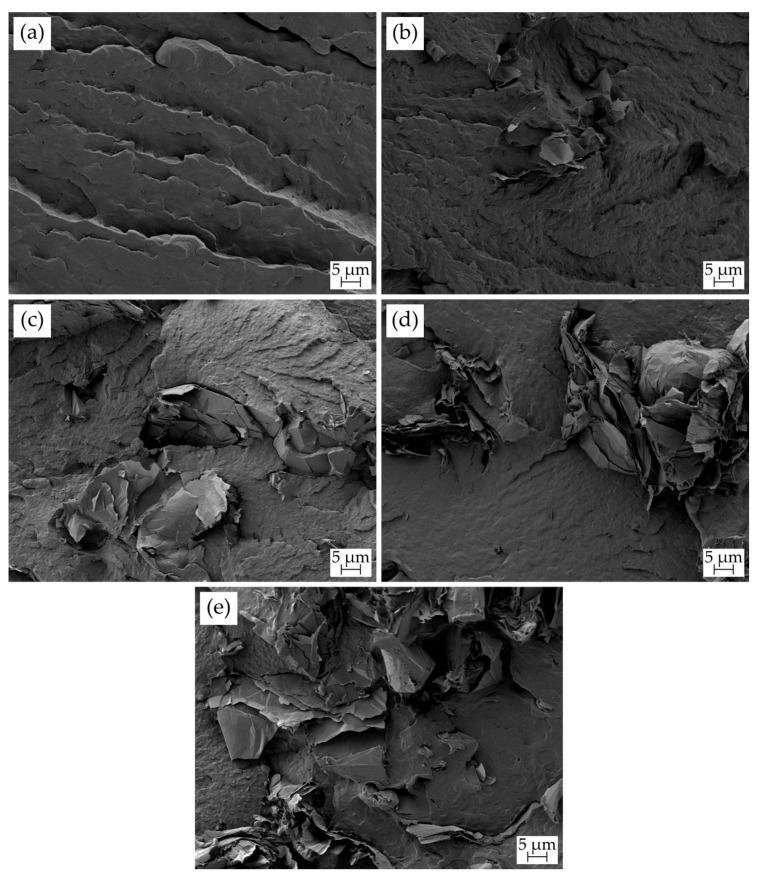
Field emission scanning electron microscopy (FESEM) images at 1000× of the fracture surfaces of the PA1001/EGr composites: (**a**) PA1010; (**b**) PA1010/2.5EGr; (**c**) PA1010/5EGr; (**d**) PA1010/7.5EGr; (**e**) PA1010/10EGr.

**Figure 3 polymers-14-01843-f003:**
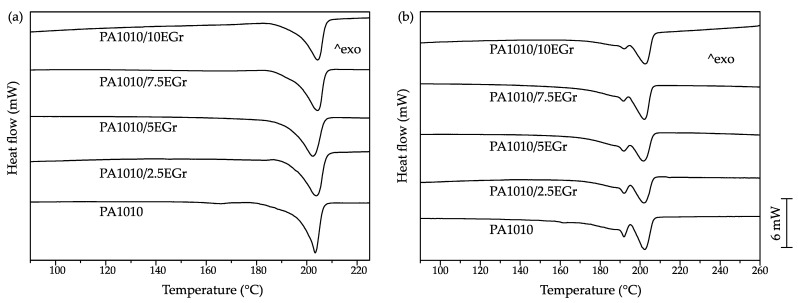
Differential scanning calorimetry (DSC) thermograms of PA1010/EGr composites, (**a**) first heating cycle and (**b**) second heating cycle.

**Figure 4 polymers-14-01843-f004:**
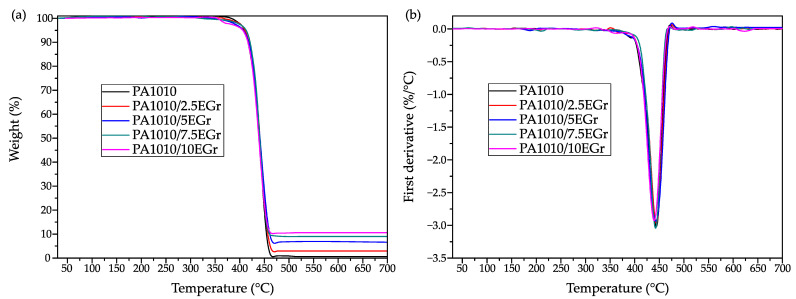
Thermal degradation of PA1010/EGr composites, (**a**) thermogravimetric (TGA) curves and (**b**) first derivative (DTG) curves.

**Figure 5 polymers-14-01843-f005:**
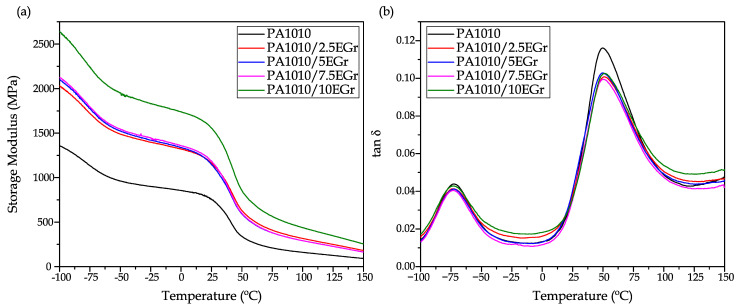
Thermomechanical properties of PA1010/EGr composites as a function of temperature, (**a**) storage modulus (E’) and (**b**) dynamic damping factor (tan δ).

**Figure 6 polymers-14-01843-f006:**
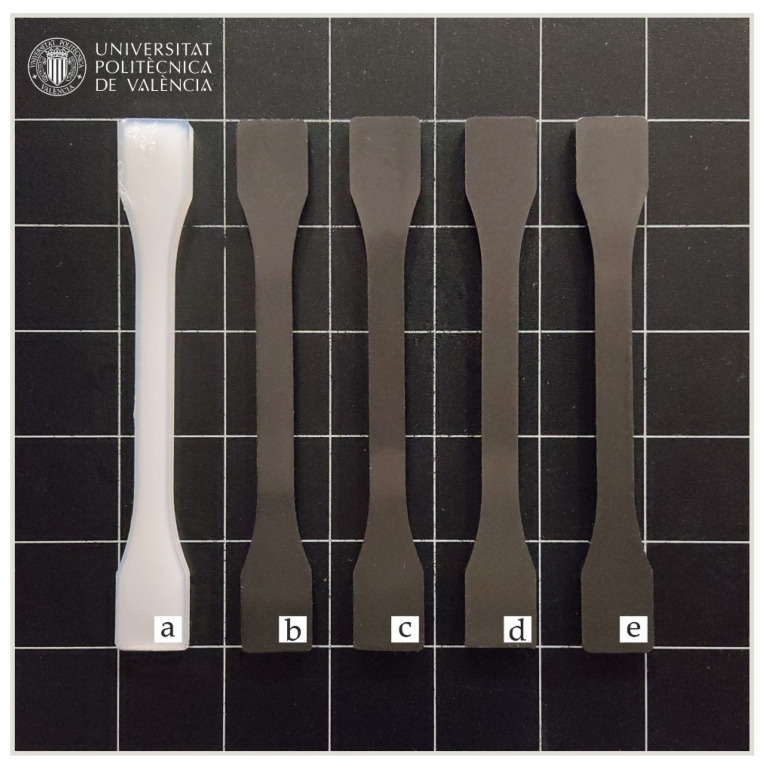
Visual appearance of the samples: (**a**) PA1010; (**b**) PA1010/2.5EGr; (**c**) PA1010/5EGr; (**d**) PA1010/7.5EGr; (**e**) PA1010/10EGr.

**Figure 7 polymers-14-01843-f007:**
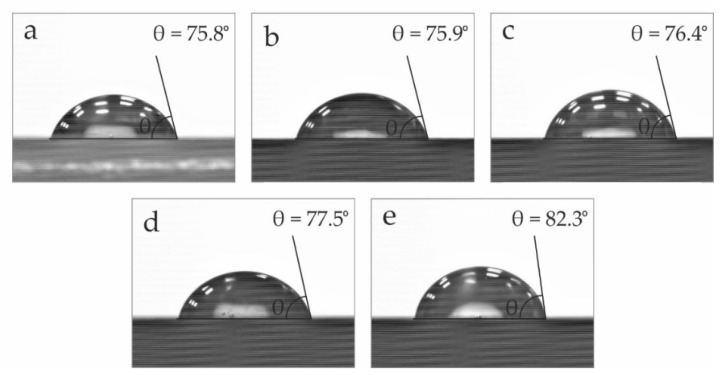
Water contact angles of the different composites: (**a**) PA1010; (**b**) PA1010/2.5EGr; (**c**) PA1010/5EGr; (**d**) PA1010/7.5EGr; (**e**) PA1010/10EGr.

**Figure 8 polymers-14-01843-f008:**
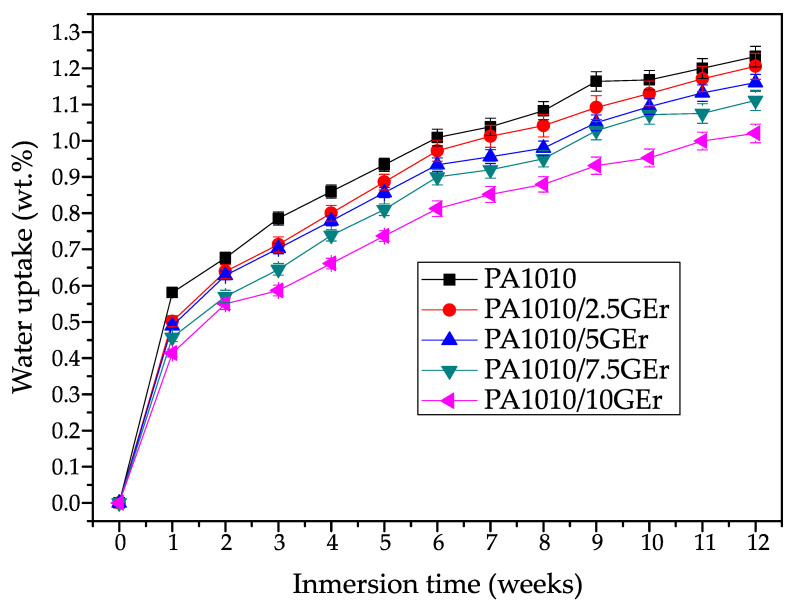
Water uptake of PA1010/EGr composites.

**Figure 9 polymers-14-01843-f009:**
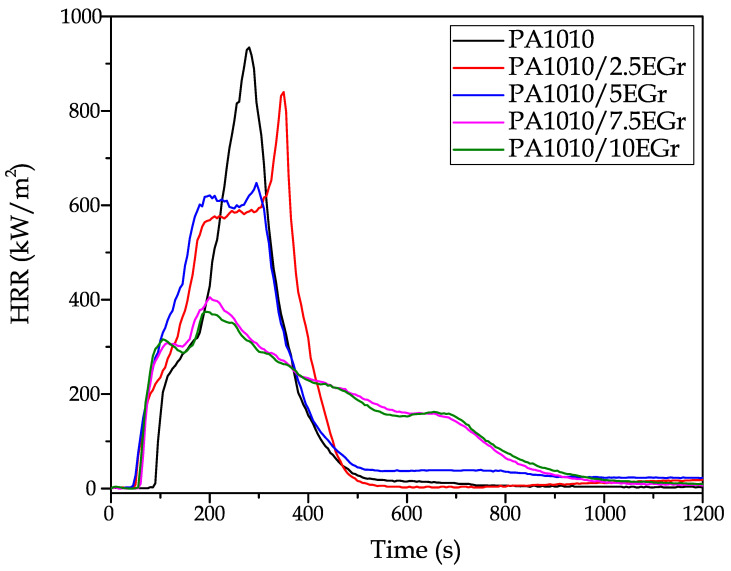
HRR evolution of the PA1010/EGr composites.

**Figure 10 polymers-14-01843-f010:**
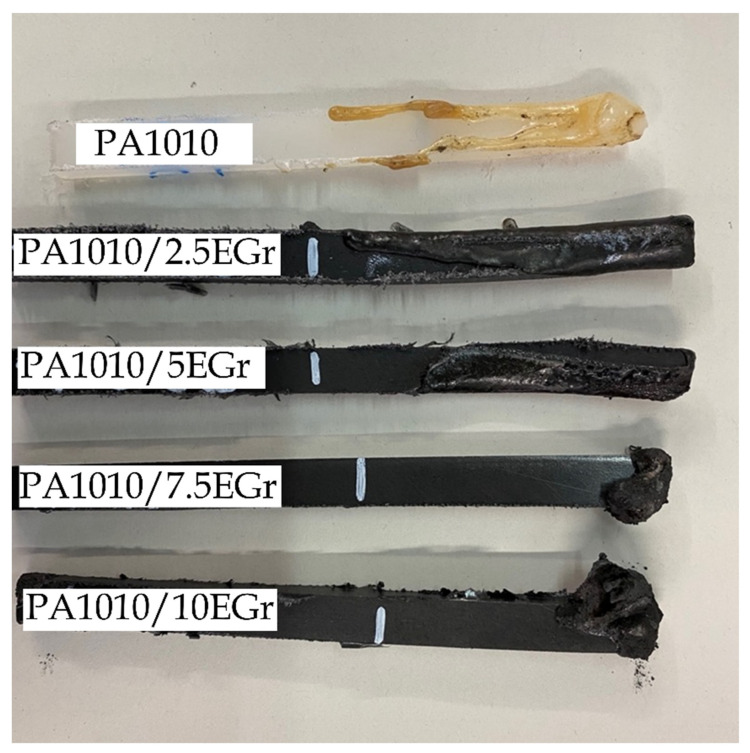
Visual appearance of the samples after UL-94 test.

**Table 1 polymers-14-01843-t001:** Summary of compositions according to the weight content (wt.%) of PA1010 and different proportions of expandable graphite (EGr).

Code	PA1010 (wt.%)	EGr (wt.%)
PA1010	100	0
PA1010/2.5EGr	97.5	2.5
PA1010/5EGr	95	5
PA1010/7.5EGr	92.5	7.5
PA1010/10EGr	90	10

**Table 2 polymers-14-01843-t002:** Summary of mechanical properties of the PA1010/EGr composites in terms of: tensile modulus (E), maximum tensile strength (σ_max_), elongation at break (ε_b_), Shore D hardness and impact strength.

Code	E (MPa)	σ_max_ (MPa)	ε_b_ (%)	Shore D Hardness	Impact Strength (kJ/m^2^)
PA1010	1701 ± 5	45.5 ± 0.8	237.4 ± 11.3	74.2 ± 0.3	9.5 ± 0.2
PA1010/2.5EGr	2046 ± 8	50.5 ± 0.5	18.1 ± 0.7	74.6 ± 0.3	5.1 ± 0.2
PA1010/5EGr	2067 ± 22	48.4 ± 0.3	13.3 ± 0.3	75.1 ± 0.2	4.5 ± 0.2
PA1010/7.5EGr	2085 ± 12	46.5 ± 0.7	6.9 ± 0.4	75.3 ± 0.2	2.5 ± 0.3
PA1010/10EGr	2164 ± 15	42.9 ± 0.7	5.7 ± 0.2	76.1 ± 0.3	2.0 ± 0.2

**Table 3 polymers-14-01843-t003:** Thermal parameters of the composites with different amounts of expandable graphite in terms of: melting temperature (T_m_), normalized melting enthalpy (ΔH_m_) and degree of crystallinity (X_c_).

Samples	First Heating			Second Heating			
	T_m_ (°C)	ΔH_m_ (J/g)	X_c_ (%)	T_m1_ (°C)	T_m2_ (°C)	ΔH_m_ (J/g)	X_c_ (%)
PA1010	203.4 ± 0.3	49.5 ± 0.4	20.3 ± 0.3	192.1 ± 0.2	202.3 ± 0.2	46.7 ± 0.4	19.1 ± 0.3
PA1010/2.5EGr	203.7 ± 0.3	46.1 ± 0.5	19.4 ± 0.4	192.2 ± 0.1	202.0 ± 0.1	46.0 ± 0.5	19.3 ± 0.4
PA1010/5EGr	202.4 ± 0.4	46.0 ± 0.8	19.8 ± 0.5	192.0 ± 0.2	201.7 ± 0.1	48.2 ± 0.8	20.8 ± 0.5
PA1010/7.5EGr	204.2 ± 0.2	46.1 ± 0.7	20.4 ± 0.4	191.7 ± 0.3	202.1 ± 0.3	49.0 ± 0.7	21.7 ± 0.4
PA1010/10EGr	204.3 ± 0.3	46.5 ± 0.6	21.2 ± 0.4	192.1 ± 0.2	202.6 ± 0.2	47.5 ± 0.6	21.6 ± 0.4

**Table 4 polymers-14-01843-t004:** Main thermal degradation parameters of the composites with different amounts of expandable graphite in terms of: initial temperature of degradation at mass loss of 5% (T_5%_), maximum degradation rate temperature (T_deg_) and residual mass at 700 °C.

Samples	T_5%_ (°C)	T_deg_ (°C)	Residual Weight (%)
PA1010	409.1 ± 1.4	444.3 ± 1.0	0.5 ± 0.1
PA1010/2.5EGr	410.6 ± 1.1	443.5 ± 0.9	3.2 ± 0.1
PA1010/5EGr	407.4 ± 1.2	443.7 ± 0.6	6.8 ± 0.3
PA1010/7.5EGr	412.8 ± 1.6	442.3 ± 1.0	9.2 ± 0.4
PA1010/10EGr	406.1 ± 1.3	438.9 ± 0.8	10.5 ± 0.3

**Table 5 polymers-14-01843-t005:** Thermomechanical properties of PA1010/EGr composites obtained via dynamic mechanical thermal analysis (DMTA).

Samples	E’ (MPa) at −90 °C	E’ (MPa) at 0 °C	E’ (MPa) at 125 °C	T_g_ (°C)
PA1010	1285 ± 42	850 ± 15	125 ± 4	49.7 ± 0.9
PA1010/2.5EGr	1925 ± 55	1320 ± 25	250 ± 7	51.2± 0.7
PA1010/5EGr	1990 ± 50	1340 ± 17	230 ± 6	49.9 ± 1.1
PA1010/7.5EGr	2015 ± 60	1360 ± 36	230 ± 10	50.4 ± 0.9
PA1010/10EGr	2500 ± 75	1740 ± 34	350 ± 8	51.7 ± 1.2

**Table 6 polymers-14-01843-t006:** Luminance and colour coordinates (L*, a*, b* and C*, h*) of PA1010/EGr samples.

Code	L*	a*	b*	C*	h*
PA1010	62.54 ± 0.11	−2.35 ± 0.06	−6.63 ± 0.05	7.03 ± 0.29	250.47 ± 8.36
PA1010/2.5EGr	36.22 ± 0.04	0.22 ± 0.12	0.66 ± 0.07	0.69 ± 0.05	71.66 ± 3.55
PA1010/5EGr	36.08 ± 0.05	0.22 ± 0.08	0.64 ± 0.09	0.68 ± 0.10	71.12 ± 3.12
PA1010/7.5EGr	35.82 ± 0.03	0.23 ± 0.05	0.72 ± 0.08	0.75 ± 0.08	72.37 ± 4.36
PA1010/10EGr	34.92 ± 0.05	0.24 ± 0.03	0.75 ± 0.09	0.85 ± 0.12	72.34 ± 2.95

**Table 7 polymers-14-01843-t007:** Summary of thermal parameters obtained with the calorimetric cone test (CCT) on the PA1010 and EGr samples.

Code	TTI (s)	t_sos.inflamability_ (s)	pHRR (kW/m^2^)	tpHRR (s)	EHC (MJ/kg)	THR (MJ/m^2^)	FRI
PA1010	82 ± 3	811 ± 9	934 ± 15	280 ± 4	31.5 ± 1.9	155.5 ± 4.2	1
PA1010/2.5EGr	43 ± 2	510 ± 4	840 ± 9	350 ± 5	34.1 ± 1.6	184.2 ± 5.3	0.49
PA1010/5EGr	42 ± 2	918 ± 12	647 ± 12	295 ± 4	35.6 ± 2.1	185.7 ± 4.7	0.60
PA1010/7.5EGr	53 ± 3	1029 ± 15	405 ± 9	200 ± 4	31.0 ± 1.4	176.5 ± 2.6	1.31
PA1010/10EGr	50 ± 2	987 ± 9	374 ± 6	195 ± 3	31.6 ± 1.2	176.4 ± 3.7	1.34

**Table 8 polymers-14-01843-t008:** Smoke parameters obtained with CCT on the PA1010 and EGr samples.

Code	SEA (m^2^/kg)	CO_2_ Yield_max_ (kg/kg)	CO Yield_max_ (kg/kg)	Total Smoke (m^2^/m^2^)
PA1010	545 ± 5.6	2.80 ± 0.12	0.031 ± 0.002	1050.3 ± 56.5
PA1010/2.5EGr	362.3 ± 4.8	3.08 ± 0.14	0.033 ± 0.001	1176.3 ± 45.5
PA1010/5EGr	541.4 ± 8.9	2.74 ± 0.09	0.032 ± 0.003	1202.0 ± 75.1
PA1010/7.5EGr	317.7 ± 6.5	2.78 ± 0.10	0.066 ± 0.004	1178.5 ± 63.3
PA1010/10EGr	266.2 ± 5.9	2.83 ± 0.11	0.061 ± 0.003	1030.5 ± 57.8

**Table 9 polymers-14-01843-t009:** Smoke parameters obtained with CCT on the PA1010 and EGr samples.

Code	LOI (% O_2_)	UL 94
PA1010	24.2 ± 5.6	V-2
PA1010/2.5EGr	25.2 ± 4.8	V-2
PA1010/5EGr	23.6 ± 8.9	V-2
PA1010/7.5EGr	22.6 ± 6.5	V-1
PA1010/10EGr	22.1 ± 5.9	V-1

## Data Availability

Not applicable.

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
