# Peer review of "Development and Characterization of High Environmentally Friendly Composites of Bio-Based Polyamide 1010 with Enhanced Fire Retardancy Properties by Expandable Graphite"

_polymers, 2022, doi:10.3390/polym14091843_

Round 1

Reviewer 1 Report

The research topic of the manuscript "Development and characterization of high environmentally friendly composites of bio-based polyamide 1010 with enhanced fire retardancy properties by expandable graphite" is interesting, and the used experimental techniques are appropriate.

The Abstract is well prepared, presenting the main novelty in the research and the main results achieved.

Please give "Expandable graphite" and "Mechanical properties" in lower case for keywords.

Overall, the Introduction is well-constructed, but it would be good to highlight the novelty and the need for research. At the end of this part, add another paragraph about the study's originality and difference from the research already done and cited in the text.

In Materials and Methods: Figure 1a) is relatively low quality (quite blurred). Therefore, I would ask the authors to replace it with a better one.

The data in Table 1 "Summary of composition…" need additional justification, and that is, it must be justified why the authors chose these values. That can, of course, be done by quoting previous research or referring to preliminary experiments.

In the Results - Figure 5 (lines 418-419) is somewhat blurry, so please correct it. The same is valid for Figure 9 (lines 588-589).

There is no conclusion in the manuscript. My strong recommendation is for part three to become "Results and discussion" and part four to be transformed into "Conclusions". In the conclusion of the manuscript, the main results achieved in the study must be presented much more clearly, and hence the authors should point out the main contributions of this study.

The References cited are appropriate.

Author Response

Dear reviewer, we highly appreciate your rapid response. Please review the attached document with the answers to your suggestions.

Reviewer 2 Report

The manuscript "Development and characterization of high environmentally friendly composites of bio-based polyamide 1010 with enhanced fire retardancy properties by expandable graphite" shows an interesting concept of obtaining flame retardant in PA regarding different percentage of EG. The script is well written and well readable also for non-experts in this topic.

There are some minor parts need to be addressed.

1.In general the abbreviation of expandable graphene in EG might be misleading to ethylene glycol which is expressed in same term. Maybe change to EGr instead to avoid confusion.

2. How did the authors verify a good homogenisation of EG in PA hence EG are particles and might concentrate to the bottom. Please comment such

3.The authors did different percentage with results in view of color change as expected became basically black. There is that concern especially for certain plastics which needs transparency in their applications. Therefore please address those and are those EG applied in real products already? 

4. One part the authors didn't mention hence EG is known being thermal conductive and for some thermoplastic polymers those properties might effect their application range.

5. Last at least it might be beneficial to add a Table in the discussion to show where such EG additives can be applied using references from other works and what other applications are aimed for. 

Author Response

(The authors gave the same response as above.)

Round 2

Reviewer 1 Report

The esteemed authors have complied with or explained all my recommendations and comments. That has increased the manuscript's quality and given me a reason to recommend accepting the manuscript in its present form.